# *Slaapte* or *Sliep*? Extending Neural-Network Simulations of English Past Tense Learning to Dutch and German

**Xiulin Yang**  **Jingyan Chen**  **Arjan van Eerden**

**Ahnaf Mozib Samin**  **Arianna Bisazza**

`{x.yang.31, j.chen.63, a.j.van.eerden, a.m.samin}@student.rug.nl`
`a.bisazza@rug.nl`
University of Groningen, The Netherlands

## Abstract

This work studies the plausibility of sequence-to-sequence neural networks as models of morphological acquisition by humans. We replicate the findings of Kirov and Cotterell (2018) on the well-known challenge of the English past tense and examine their generalizability to two related but morphologically richer languages, namely Dutch and German. Using a new dataset of English/Dutch/German (ir)regular verb forms, we show that the major findings of Kirov and Cotterell (2018) hold for all three languages, including the observation of over-regularization errors and micro U-shape learning trajectories. At the same time, we observe troublesome cases of non human-like errors similar to those reported by recent follow-up studies with different languages or neural architectures. Finally, we study the possibility of switching to orthographic input in the absence of pronunciation information and show this can have a non-negligible impact on the simulation results, with possibly misleading findings.

## 1 Introduction

The plausibility of neural network-based or connectionist models in simulating psycholinguistic behaviours has been attracting considerable attention since Rumelhart and McClelland (1986) first modeled the past-tense acquisition with an early example of sequence-to-sequence network. Their experiment received harsh criticism (e.g., Pinker and Prince, 1988) but also inspired cognitive scientists with alternatives (e.g., Kirov and Cotterell, 2018; Plunkett and Juola, 1999; Taatgen and Anderson, 2002). Much more recently, Kirov and Cotterell (2018) replicated Rumelhart

and McClelland (1986)'s simulations using a modern encoder-decoder neural architecture developed for the task of morphological paradigm completion. Their improved results resolved much of the original criticisms by Pinker and Prince (1988).

The main purpose of this paper is to study the generalizability of Kirov and Cotterell (2018)'s findings beyond the case of English. Specifically, we consider two languages that are genetically related to English, but morphologically richer – namely, Dutch and German. In these languages too, past tense inflection is divided into regular and irregular verbs, but with different proportions and different inflectional patterns than English. Moreover, German and Dutch are characterized by a much more transparent orthography than English (Van den Bosch et al., 1994; Marjou, 2021), which allows us to study the usability of grapheme-based input for simulating past tense acquisition patterns when pronunciation information may not available. Concretely, we aim to answer the following research questions:

1. Can the model applied by Kirov and Cotterell (2018) to English also simulate the past tense acquisition process in languages with more complex morphological inflection, such as Dutch and German?

2. Given the more predictable grapheme-to-phoneme correspondence, i.e., orthographic transparency (Marjou, 2021), in these two languages, will the model perform similarly if the written forms of verbs are used for training instead of the phonetic ones?

To answer these two questions, we build and release a new past-tense inflection dataset of English, Dutch, and German, covering both grapheme and phoneme features (Section 3).[1] We

---

[1] All code and data are available at `https://github.com/JingyanChen22/IK-NLP-Project-4.git`

then replicate the single-task learning experiments of Kirov and Cotterell (2018) (Section 4) and extend them to our multilingual dataset, using both phoneme- and grapheme-based input for comparison (Section 5).

Our findings reconfirm the potential and limitations of using neural networks for the simulation of human language learning patterns. Our model shows human-like behavior in learning past tenses of verbs, such as the micro U-shape coined by Plunkett et al. (1991) and over-regularization errors in all the examined languages; however non human-like errors are also reported. We also find that learning irregular past tense forms is considerably easier in Dutch and German than in English. Finally, we observe that higher orthographic transparency indeed leads to more consistent learning results when a model is trained with grapheme vs. phoneme input.

## 2 Background

**Past tense debate**   The acquisition of verbal past tense in English, particularly the over-regularization of the irregular verbs in the process of learning (Marcus et al., 1992), has been serving as a testing ground for different hypotheses in language modelling for decades. A much debated question is whether the past tense of (ir)regular verbs is learnt by rules and memories (e.g., Plaut and Gonnerman, 2000; Seidenberg and Gonnerman, 2000; Marcus et al., 1995; Albright and Hayes, 2003; Pinker and Ullman, 2002), by analogy (e.g., Ramscar, 2002; Albright and Hayes, 2003) or by a dual mechanism (Pinker and Prince, 1988; Taatgen and Anderson, 2002).

Marcus et al. (1995) posited the necessity of mental rules in learning German irregular verbs. By contrast, Ernestus and Baayen's (2004) and Hahn and Nakisa's (2000) studies on Dutch and German respectively provided evidence in favour of connectionist and analogical approaches: they showed that humans tend to choose wrong past tense suffixes for regular verbs whose phonological structure is similar to that of irregular ones.

**Recent connectionist *revival***   The recent development of deep learning methods in computational linguistics has led to a renewed interest in connectionist approaches to modelling language acquisition and processing by humans (e.g., Blything et al., 2018; Kádár et al., 2017; Pater, 2019; Corkery et al., 2019; McCurdy et al., 2020). Last

year, modelling morphological acquisition trajectories was adopted as one of the shared tasks of SIGMORPHON-UniMorph (Kodner and Khalifa, 2022). The three submitted neural systems (Pimentel et al., 2021; Kakolu Ramarao et al., 2022; Elsner and Court, 2022) exhibited over-regularization and developmental regression, but non-human-like behaviours were also observed.

Some recent studies have revealed a poor alignment between the way humans and neural encoder-decoder models generalize to new words (*wug* test) in the case of English verb past tense (Corkery et al., 2019) and German plural nouns (McCurdy et al., 2020). Dankers et al. (2021) observed cognitively plausible representations in a recurrent neural network (RNN) trained to inflect German plural nouns but also found evidence of problematic 'shortcut' learning. Wiemerslage et al. (2022) observed that Transformers resemble humans in learning the morphological inflection of English and German in the *wug* tests but they also pointed out the divergence of the model in German production. However, computational simulations have succeeded in replicating the U-shaped learning curve during the acquisition of past tense (Kirov and Cotterell, 2018; Plunkett and Marchman, 2020). Additionally, further probing experiments have suggested that neural models do learn linguistic representations (Goodwin et al., 2020; Hupkes et al., 2018; Ravichander et al., 2020). Our research continues on exploring the cognitive plausibility of neural networks in modeling language inflection learning.

**Recurrent encoder-decoder inflection model** In this work, we adopt the model of Kirov and Cotterell (2018), henceforth referred to as **K&C**. This model is based on the encoder-decoder architecture proposed by Bahdanau et al. (2014), with input representation and hyper-parameters taken from Kann and Schütze (2016). The architecture consists of a bidirectional LSTM (BiLSTM) encoder augmented with an attention mechanism and a unidirectional LSTM decoder. The task of the encoder is to map each phonetic (or orthographic) symbol from the input string to a unique embedding and then process that embedding to get a context-sensitive representation of that symbol. The decoder reads the context vector from the final cell of the encoder and generates an output of phoneme/grapheme sequences through training a BiLSTM model with two hidden layers. For

more details on the model, see Bahdanau et al. (2014); Kann and Schütze (2016); Kirov and Cotterell (2018).

## 3 Datasets

To replicate the results published by K&C, we employ their dataset based on CELEX (Baayen et al., 1993).[2] To extend the experiments to Dutch and German and compare the results to English, we build a new dataset containing past tense forms in all three languages.

### 3.1 K&C English Dataset

K&C's CELEX-based dataset contains 4,039 English verb types including 3,871 regular verbs and 168 irregular verbs. Each verb is associated with an infinitive form and past tense form, both in International Phonetic Alphabet (IPA). Moreover, each verb is marked as regular or irregular (Albright and Hayes, 2003).

Note that there are label errors in their dataset. For example, `dive-dived`, `dream-dreamed`, `light-lighted` are marked as *irregular*. This is possibly because those verbs have two past tense forms and the other form does not follow the regular inflection (`dive-dove, dream-dreamt, light-light`). However, as the past tense of those verbs in the original dataset aligns with the regular inflection rule of English, we take those verbs as regular ones and manually correct their labels.

### 3.2 Multilingual Unimorph-based Dataset

We use the morphological annotation dataset Unimorph (McCarthy et al., 2020) as a source of English, Dutch, and German word forms to enable a fair comparison in our multilingual experiments. In this lexicon, each entry consists of the infinitive of the verb, the conjugation, and the tag containing the Part-Of-Speech and inflectional information. Our use of the Unimorph dataset allowed for a wider range of past tense inflection cases compared to the CELEX-based dataset. Unlike the latter, we included more present-past pairs instead of exclusively using infinitive-past pairs. An important adjustment has to be made here because English has only two forms for the present tense (*I/you/we/they*) and only one for the past. By contrast, Dutch and German distinguish more persons

[2]Dataset, code and other experimental details are taken from `https://github.com/ckirov/RevisitPinkerAndPrince`

| present(g) | past(g) | present(p) | past(p) | reg |
|---|---|---|---|---|
| accounts | accounted | @k6nts | @k6ntId | reg |
| account | accounted | @k6nt | @k6ntId | reg |
| feels | felt | filz | fElt | irreg |
| feel | felt | fil | fElt | irreg |

(a) English

| | | | | |
|---|---|---|---|---|
| slaap | sliep | slap | slip | irreg |
| slaapt | sliep | slapt | slip | irreg |
| slapen | sliepen | slap@ | slip@ | irreg |
| behoef | behoefde | b@huf | b@huvd@ | reg |
| behoeft | behoefde | b@huft | b@huvd@ | reg |
| behoeven | behoefden | b@huv@ | b@huvd@ | reg |

(b) Dutch

| | | | | |
|---|---|---|---|---|
| berechne | berechnete | b@rExn@ | b@rExn@t@ | reg |
| berechnest | berechnetest | b@rExn@st | b@rExn@t@st | reg |
| berechnet | berechnete | b@rExn@t | b@rExn@t@ | reg |
| berechnen | berechneten | b@rExn@n | b@rExn@t@n | reg |
| fliehe | floh | fli@ | flo | irreg |
| fliehst | flohst | flist | flost | irreg |
| flieht | floh | flit | flo | irreg |
| fliehen | flohen | fli@n | flo@n | irreg |

(c) German

Figure 1: Excerpt of the newly introduced dataset of English, Dutch and German past tense. Dutch verbs: `slapen` (*to sleep*); `behoeven` (*to need*). German: `berechnen` (*to calculate*); `fliehen` (*to fleed*).

in both present and past tense. To address this, we include for each lemma the first/second/third singular present form and plural form together with their respective past form, each as a separate entry (see examples in Figure 1).

Specifically, we start by extracting from Unimorph a list of verb lemmas and their corresponding present and past tense forms. A different extraction script is used in each language because of the different number of forms and slightly different POS tags:

- English only has two present tense forms: one for the third person singular and one for the rest. Mostly, there is only one past tense.

- Most verbs in Dutch have three present tense forms and two past tense forms.

- Most verbs in German have five present tense forms and four past tense forms.

Next, we tag each form as regular or irregular, based on a simple rule-based strategy:

- English: if the past tense ends with 'ed' then it is considered a regular verb.

- Dutch: if the singular past tense ends with '-de' or 'te', it is considered regular.

| | | Number of verbs | | | | | | | |
|---|---|---|---|---|---|---|---|---|---|
| **Language** | **Type** | **train** | | **dev** | | **test** | | **Total verbs** | |
| | | **Count** | **(%)** | **Count** | **(%)** | **Count** | **(%)** | **Count** | **(%)** |
| English | all | 4,879 | 79.9 | 611 | 10.0 | 614 | 10.1 | 6,104 | 100.0 |
| | regular | 4,601 | 75.4 | 529 | 8.7 | 520 | 8.5 | 5,650 | 92.6 |
| | irregular | 278 | 4.6 | 82 | 1.3 | 94 | 1.5 | 454 | 7.4 |
| Dutch | all | 4,896 | 80.1 | 612 | 10.0 | 607 | 9.9 | 6,115 | 100.0 |
| | regular | 4,383 | 71.7 | 550 | 9.0 | 542 | 8.9 | 5,475 | 89.6 |
| | irregular | 513 | 8.4 | 62 | 1.0 | 65 | 1.0 | 640 | 10.4 |
| German | all | 4,865 | 79.7 | 616 | 10.1 | 620 | 10.2 | 6,101 | 100.0 |
| | regular | 4,299 | 70.5 | 535 | 8.8 | 578 | 9.5 | 5,412 | 88.8 |
| | irregular | 566 | 9.2 | 81 | 1.3 | 42 | 0.7 | 689 | 11.2 |

Table 1: Dataset distributed into train, dev and test sets in each of the three languages. The number of regular and irregular verbs is also reported. The percentage is calculated over the total number of verbs per language.

- German: if the singular past tense of the first or third person ends with '-te', it is considered regular.

Finally, the IPA transcriptions of all word forms are retrieved from CELEX for all languages and added to the final dataset. As shown in Figure 1, the resulting dataset is in the same format as K&C's CELEX-based dataset.

**Data selection** The generated Dutch data only contains 6106 verb forms *versus* 11489 and 6975 in English and German respectively. Therefore, to enable a fair comparison among languages, we need to downsample the larger datasets. However, randomly choosing 6K verb forms from the English and German lists may lead to a poor selection given the long tail of infrequent words. As a solution, we use word form frequencies as provided in the CELEX data and choose *all* words with a frequency of more than 1 in a million, and complement with a random selection of less frequent words in order to get approximately 6106 verb forms.

To make sure the model can generalize to unseen verbs, we follow Goldman et al. (2022) and split the data by lemma into a train set (80%), a development (dev) set (10%) and a test set (10%). Therefore, the verb forms from the same lemma can only appear in one of the splits. The data distribution into three sets and regular/irregular verbs for each language is reported in Table 1.

### 3.3 Remarkable problems

A few problems occurred during data preparation. First, rule-based tagging of lemma's is not as trivial as it seems at first sights. For example, in English, not all past tenses ending with '-ed' are regular. Using the data of K&C, we added a few exceptions that are all irregular words ending with '-ed': `bled`, `bred`, `led`, `misled`, `fled`, and forms of `fed` (including `breast-fed`, `force-fed` and `bottle-fed`).

Also, in the original K&C experiment, the model should be able to predict past tense based on what it learned from other verbs, not from other word forms. In morphologically richer languages, a lemma has more word forms and data splitting becomes problematic. For instance, a model might have learned that `work` → `worked` and `walks` → `walked`, then it might predict that `works` → `worked`. In such a case, it is not possible to know whether the model made the right prediction based on similarities to other lemmas (`walks`) or to other forms of the same verb (`work`). To be as comparable as possible to the original setup of K&C, we put all forms of the same verb in the same data split (that is, either training, dev or test). As a result, if the model scores well, we know for sure that it cannot make predictions based on other forms of the same verb.

Another issue is that one present tense form normally corresponds to one past tense form. However, German poses two notable exceptions to this:

- The second person singular verb form ends with '-st' and the third person singular ends with '-t'. Those forms coincide if a verb already ends with an 's', but there is still a difference between those forms in the past tense. For example, `bremst` is the present conju-

gation form of verb `bremsen` (*to brake*) for pronoun `du` *you*, `er` *he* and even `ihr` *you*.

- Verbs ending in '-t' can be the third person singular or the second person plural informal. For example, `wundert` is the present conjugation of the verb `wundern` (*to wonder*) for the pronoun `ihr` *you* and `er` *he*.

In the former case, the model should be able to output multiple solutions, since only context can make clear whether it is the second person or the third person. However, this complicates the evaluation. As a solution, we exclude the third person form if it collides with the second person. As for the latter issue, we choose to remove all second person plural informal forms, since those are far less frequent than the third person singular forms.

## 4 Replication of K&C

Before moving to the main multilingual experiments, we replicate the original K&C experiments (single-task only).

### 4.1 Experimental Setup

For the replication, we employ K&C's CELEX-based dataset and keep the model architecture and hyper-parameters unchanged using Open-NMT (Klein et al., 2017)[3]. Also, as reported by K&C, we train the neural model for 100 epochs to make sure the examples in the training data are properly learned. See more details in Appendix A. Following K&C, the model is trained on the IPA transcription.

We use word form-level accuracy to evaluate model performance. An important remark concerns data splitting: K&C did not release their specific data split, which makes it impossible to replicate the exact same results. We, therefore, create our own splits following K&C's proportions (80/10/10% for training/dev/test). To obtain more reliable results, we train the model three times using different random seeds for different initialization and report the averaged resulting accuracies.

To study the micro U-shape learning curve of irregular verbs, we save the model at each 10 epochs and use those partially-trained models to predict the test set and compare their prediction results.

---

[3]However, as the epoch has been deprecated in the latest version of OpenNMT, we converted it to train_steps based on its relationship with steps.

### 4.2 Results

As shown in Table 2, the results on the training set are almost the same as reported in the original paper, which means our replication is largely successful.[4] We note that the accuracy for irregular verbs in the dev and test set is considerably different from that of K&C (dev: 21.1% vs. 53.3%; test: 35.3% vs. 28.6%). Since K&C did not release their specific data split, replicating their exact results on the small portion of irregular verbs is not possible. Given that our results are averaged over three random seeds and on all three split sets, we consider them more reliable, which means the model might perform worse at learning the past tense of irregular verbs than K&C's report.

| | all | | | regular | | | irregular | | |
|---|---|---|---|---|---|---|---|---|---|
| | train | dev | test | train | dev | test | train | dev | test |
| K&C | 99.8 | 97.4 | 95.1 | 99.9 | 99.2 | 98.9 | 97.6 | 53.3 | 28.6 |
| Ours | 99.9 | 95.3 | 96.5 | 99.9 | 98.4 | 99.2 | 98.4 | 21.1 | 35.3 |

Table 2: Mean accuracy of our replication of K&C with three random seeds based on English data from CELEX-based dataset.

### 4.3 Discussion

The reason we assume for the gap between our results and K&C's is twofold: (i) the number of irregular verbs is much lower than regular ones, which makes the accuracy change dramatically even if only few more or few less verbs are predicted correctly than the original experiments; (ii) we corrected the label errors mentioned above, thus the number of irregular verbs becoming smaller than before. This small difference could cause a large impact on the accuracy calculation given that these two sets only contain about 20 irregular verbs. To test this hypothesis, we conduct 9-fold cross-validation[5] and find that the accuracy for irregular verbs varied in different dev splits, ranging widely between 9% and 42%.

---

[4]Our results are also very close to those of Corkery et al. (2019), who did a similar replication and reported the averaged accuracy over ten runs initialized with different random seeds, but only on the training set.

[5]We keep the test set unchanged and validated across the train and dev sets. To make sure the dev set has a comparable number of verbs as the original set, we adopt 9 fold instead of 10 fold cross-validation.

# 5 Multilingual Experiments

This section presents the results of our main experiments aimed at comparing Dutch and German past learning patterns to the English ones. It also presents the results of grapheme *vs* phoneme sequence learning in all three languages. Because Dutch and German pronunciation is more predictable than the English one, we expect that the difference between grapheme and phoneme learning will be smaller in these languages.

For comparability, all experiments in this section use the newly introduced Unimorph-based dataset, which includes a similar amount of training forms in all languages (cf. Table 1). The model architecture and the hyperparameter settings are the same as in previous experiments. We also run each experiments three times with different random seeds and report the averaged results.

We use our newly-created data for multilingual experiments without resampling tokens by their frequency. This decision is informed by research suggesting that human learners generalize over type frequency, rather than token frequency (Bybee, 1995; Bybee and Thompson, 1997) and is consistent with the experimental design of K&C. Other studies have suggested that word frequency is important for children's past tense acquisition (Plunkett and Marchman, 1991; Bybee and Slobin, 1982; Ellis, 2002), but we do not examine this hypothesis in this work.

**Result overview** For the forms seen in training, the model is able to learn both regular and irregular past tense inflection with more than 95% accuracy (Table 3a), and with similar learning curves (Figure 2), which confirms and strengthens the main findings of K&C on two other languages.

Comparing Table 3a to 3b, we find that the overall trends are maintained when the model is trained on graphemes instead of phonemes (the original setup of K&C). However, a notable exception is observed: grapheme learning results in a much lower accuracy of English irregular verbs.

In the following sections, we discuss these results in more detail.

## 5.1 Past Tense Learning Results in English, Dutch, and German

**Accuracy** Looking closer at the results across languages (Table 3a), we notice that inflecting *unseen* Dutch regular verbs is slightly harder than in

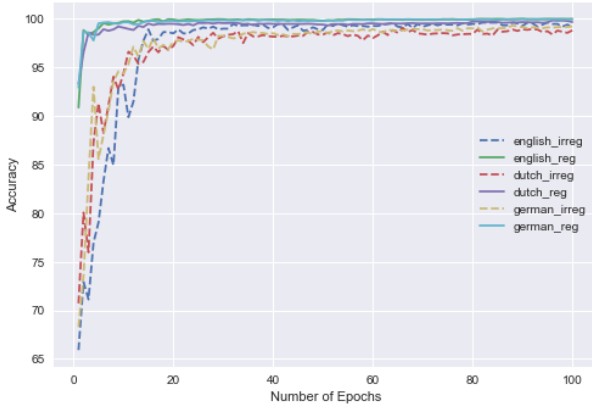

(a) Phoneme Input

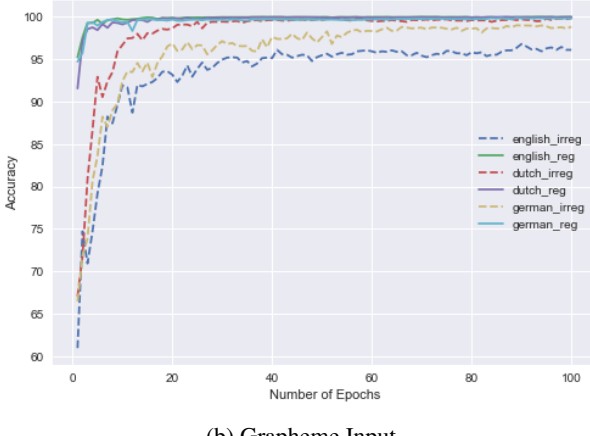

(b) Grapheme Input

Figure 2: Learning curves of the model on the German, English, and Dutch training set (with random seed *123*).

German and English. This might be explained by the fact that in Dutch all voiced consonants become unvoiced at the end of a word, but to predict if the past tense becomes '-de' (for voiced consonants) or '-te' (for unvoiced consonants), we still need the end consonant of the stem, which can be found within the lemma and most of the times in the spelling of the word form. Unfortunately, this information is absent in the pronunciation. For example, in the pair `lAnt-lAndd@`, one will not know whether the past tense should be `lAnd@` or `lAnt@` before seeing the orthographic form `land`. We find that such errors account for about 50% (18/38) of all Dutch regular verb errors. This difference in voiced/unvoiced regular past tense endings only occurs in Dutch.

As for irregular verbs, we find a large difference across languages in the ability to generalize to new forms. Especially in English, while the model has almost perfectly learned to inflect seen verbs, it has a hard time predicting the form of new irreg-

|  | all | | | regular | | | irregular | | |
|---|---|---|---|---|---|---|---|---|---|
|  | train | dev | test | train | dev | test | train | dev | test |
| EN | 99.5 | 93.1 | 92.1 | 99.8 | 96.1 | 95.0 | 98.1 | 27.8 | 40.5 |
| NL | 98.9 | 88.4 | 88.4 | 99.2 | 91.4 | 92.2 | 96.5 | 62.4 | 57.9 |
| DE | 98.9 | 85.0 | 92.5 | 99.4 | 92.0 | 95.1 | 96.7 | 38.7 | 57.9 |

(a) Phoneme input

|  | all | | | regular | | | irregular | | |
|---|---|---|---|---|---|---|---|---|---|
|  | train | dev | test | train | dev | test | train | dev | test |
| EN | 99.1 | 93.6 | 93.8 | 99.8 | 98.2 | 98.1 | 89.0 | 11.1 | 28.1 |
| NL | 99.4 | 88.0 | 89.6 | 99.8 | 91.2 | 93.0 | 97.9 | 58.6 | 61.0 |
| DE | 98.4 | 86.4 | 93.6 | 99.1 | 93.5 | 95.7 | 93.9 | 39.5 | 65.9 |

(b) Grapheme input

Table 3: Past tense inflection accuracy in English, Dutch, and German; all averaged over 3 random seeds.

| epoch | English | | Dutch | | German | |
|---|---|---|---|---|---|---|
|  | hits | | bestijgt (*mounts*) | | gilt (*applies*) | |
| 10 | hItId | hitted | b@stKGd@ | besteeg | gIlt@ | galte |
| 20 | hItst | **hit** | **b@stex** | besteeg | gIlt@ | **galt** |
| 30 | hItId | **hitted** | **b@stKGd@** | besteeg | **g&lt** | galt |
| 40 | hItId | hitted | b@stKGd@ | besteeg | g&lt | galt |
| 50 | **hIt** | hitted | b@stKGd@ | besteeg | g&lt | galt |
| 60 | **hItst** | **hit** | **b@stex** | besteeg | **gIlt@** | **gilte** |
| 70 | **hIt** | hit | b@stex | **bestijgde** | **g&lt** | **galt** |
| 80 | **hItId** | **hitted** | b@stex | **besteeg** | g&lt | galt |
| 90 | hItId | hitted | b@stex | besteeg | g&lt | galt |
| 100 | hIt | hit | b@stex | besteeg | g&lt | galt |

Table 4: The oscillating development (micro U-shape) of single verbs in three languages: with phoneme or grapheme inputs, the respectively predicted past phonetic (left) or orthographic (right) forms are changing with the training proceeding, but their final predictions are correct when reaching the last epoch. The changing points are boldfaced.

ular verbs (dev: 27.8%, test: 40.5%). This effect is smaller in Dutch and German, suggesting the irregular inflection patterns in these languages are more predictable. Surprisingly, the model made more mistakes when predicting the inflections of the irregular verbs in the German dev set than the test set (dev: 38.7%, test: 57.9%). By inspecting the mistakes, we found that the model incorrectly took many irregular verbs as regular ones because of their resemblance (high character overlap). For instance, `reitest-*reitetest/rittest` (*ride*) is influenced by the regular conjugation of `bereitest-bereitetest` (*prepare*). We found 23/81 irregular verbs in the dev set are very similar to regular verbs in the training set. Out of these, 8 irregular verbs are identical to regular ones except for a prefix (e.g., `reitet` (*rides*) vs. `bereitet` (*prepares*) and `reitest` (*ride*) vs. `verbreitest` (*spread*), which could be highly confusing for a model that is only based on form regardless of meaning. By contrast, such overlap is not found between the irregular verbs in the test set and regular ones in the training set. This distributional discrepancy might explain the lower accuracy in the dev set. It echoes with our other finding discussed in the next section that irregular verbs might be misled by regular verbs if they share representation similarity.

**Errors and learning trajectories** Going beyond overall accuracy, we inspect the learning trajectories of individual verbs in our dataset. We find human-like overregularization patterns similar to those observed by K&C in English also occur in Dutch and German. For example, in Dutch, after 40 epochs of training, the model change `verscheent` to `verscheen` as the past tense of `verschijnt` (*appears*). However, after 50 epochs, the model again generate the wrong form `verscheent`. After 70 epochs, the correct result is again obtained. Similar patterns are observed for `sink` in English and `streitet` (*argues*) in German. Interestingly, Plunkett and Marchman (1991); Bybee and Slobin (1982); Kuczaj II (1977) reported that children do sometimes vacillate, even within one utterance, between the correct and incorrect past tense form of the same irregular stem. All wrongly predicted irregular verbs are caused by over-regularization. In other words, no patterns like `ated` in English or `lookte` in Dutch are

found, which is consistent with humans' learning behaviour (Pinker and Prince, 1988). More examples from English, Dutch and German are listed in Table 4.

Additionally, we find cases where the model generates an irregular form for a regular verb, because of the resemblance with other (irregular) verbs. In Dutch, for example, the regular verb `versier-versierde` (*decorate-decorated*) gets incorrectly inflected as `*versoor` by resemblance to verbs like `verlies-verloor` (*lose-lost*). Similar errors also occur in German. For instance, the wrong prediction of `verfehle-*verfahl/verfehlte` (*miss-missed*) might be misled by the pair `befehlen-befahlen` (*order-ordered*), and `schweben-*schwoben/schwebten` (*float-floated*) is possibly due to its resemblance to `schieben-schoben` (*push-pushed*). Interestingly, this type of errors aligns with Ernestus and Baayen (2004)'s experiments with Dutch speakers: phonological similarity, rather than rule-based regularity, influences participants' judgments toward the inflection of verbs.

That said, the model also displays error patterns that are *not* human-like, such as copying the present form or randomly removing phonemes (or letters) from it. Similar cases of non-plausible predictions were also observed at the Sigmorphon Shared Task (Kodner and Khalifa, 2022), for instance `forgive-*forgaved/forgave` or `seek-*sougk/sought`. As also observed by Wiemerslage et al. (2022), this kind of model predictions contrasts with the behavior of human speakers, who mostly resort to generating a regular past tense when a verb is unknown.

### 5.2 Phoneme vs. Grapheme Input

Undoubtedly, using phoneme input is more principled than grapheme input when simulating human acquisition patterns. However, pronunciation information is not always available and makes it harder to extend this kind of simulations beyond a small set of widely studied languages. Here, we investigate the usability of grapheme-based input for modeling past tense inflection. We expect German and Dutch to be a good use case for this, given their more transparent orthography compared to English (Marjou, 2021).

The results in Table 3 clearly show that switching to grapheme input for the English simulations is not principled as this results in a slight *increase* of regular inflection accuracy (from 99.8/96.1/95.0% to 99.8/98.2/98.1% train/dev/test) as opposed to a large *decrease* of irregular inflection accuracy (from 98.1/27.8/40.5% to 89.0/11.1/28.1%). The latter effect is particularly marked, suggesting non-transparent orthography may not be a uniform property of the language but may be correlating with less regular word forms within a language. We leave this investigation to future work.

Using grapheme input in Dutch and German seems much safer (differences are overall small, with only a slight increase in almost all cases). Our observations seem to reflect the figures of Marjou (2021), who give a much higher transparency score to Dutch and German than to English.

In sum, using graphemes to simulate human patterns of morphological acquisition is possible but should be done with caution and only in some languages. A good practice could be to first verify that the orthographic transparency of a language is high (Marjou (2021) present results for 17 languages). When that is not possible, grapheme-based results should be at least validated against a small-scale pronunciation dataset.

## 6 Conclusions

In this work, we study the plausibility of using sequence-to-sequence neural networks for simulating human patterns of past tense acquisition. More specifically, we replicate findings by Kirov and Cotterell (2018) and examine their generalizability beyond the specific case of English, using a new dataset of English/Dutch/German (ir)regular verb forms based on Unimorph (McCarthy et al., 2020).

We show that the main findings of K&C also largely hold for Dutch and German, including over-regularization errors and the oscillating (or micro U-shape) learning trajectory of individual verb forms across training epochs. At the same time, we also observe cases of non human-like errors, for instance when the model just keeps the present form unchanged or randomly removes phonemes from it. A notable difference among our studied languages concern unseen English irregular verbs, which appear to be much harder to inflect than the Dutch and German ones. We also observe that the orthographic transparency of a language influences and possibly confounds the

model's learning performance: higher transparent orthography contributes to more reliable and consistent simulation results, but in general this aspect should be seriously considered when setting up new benchmarks of morphological acquisition.

Future work could include the construction of a nonce word benchmark in Dutch and German to enable a multi-lingual evaluation of this task (Corkery et al., 2019), as well as an in-depth investigation of the different level of irregular past inflection difficulty in our three languages.

Kirov and Cotterell (2018) provided very promising evidence for the use of modern neural networks to model the human language acquisition patterns. Our work confirms the potential of this research direction, but also raises important issues and joins recent follow-up studies (Corkery et al., 2019; Dankers et al., 2021; Kodner and Khalifa, 2022; Wiemerslage et al., 2022) that have warned against over-optimistic conclusions.

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

## A   Appendix

| Parameter | Value |
| --- | --- |
| seed | 123 |
| feat_vec_size | 300 |
| feat_merge | concat |
| rnn_type | LSTM |
| encoder_type | brnn |
| encoder_layers | 2 |
| encoder_rnn_size | 100 |
| decoder_type | rnn |
| decoder_layers | 2 |
| decoder_rnn_size | 100 |
| dropout | 0.3 |
| learning_rate_decay | 1.0 |
| learning_rate | 1.0 |
| batch_size | 20 |
| train_steps | (training sample size/ batch size)∗the number of epochs |
| beam_size | 12 |
| optim | adadelta |
| verbose | True |
| tensorboard | True |
| tensorboard_log_dir | logs |
| report_every | steps / 100 |
| log_file | directory of the log file |
| log_file_level | 20 |

A displays hyperparameter settings of the replicating experiments and the extension experiments.