# OpenReview forum: "Slaapte or  Sliep? Extending Neural-Network Simulations of English Past Tense Learning to Dutch and German"
_NoDaLiDa/2023/Conference — NoDaLiDa 2023_

### Official Review · Reviewer_T8uw · 2023-03-08
**Review: Slaapte or Sliep? Extending Neural-Network Simulations of English Past Tense Learning to Dutch and German**

**Rating:** 8
**Confidence:** 3

**Review:**

This paper explores the question of whether seq2seq models are plausible models for human-like morphological acquisition using a dataset of irregular past tense forms in English, German and Dutch. The paper aims to replicate the model of Kirov and Coterell (2018), but to analyse its performance with respect to more morphologically complex languages.

The paper presents their data set creation in a clear and thought-out manner.
Relevant paper to consider (For future work, as I've understood your split strategy is by "form" rather than "lemma"): https://arxiv.org/pdf/2108.05682.pdf

The results are presented in a clear manner, along with proper explanation of the differences between the authors models and the model they are replicating.

Based on your error analysis, did you consider balancing the forms based on character overlap in the training split? I guess this could have some interesting results for the evaluation.

Table 2: Clarify that this experiment is about English.

Table 4: What does bold indicate?

In all, the work is presented in a clear manner, with a reasonable experimental methodology and analysis.



**Paper Type:**

Long paper

---

### Official Review · Reviewer_vqEg · 2023-03-08
**Replication of prior work on past tense acquisition (or learning), which has arguable proof, and expansion to German and Dutch**

**Rating:** 6
**Confidence:** 4

**Review:**

The paper describes the replication of experiments by Kirov and Cotterell (2018) and extends them to study German and Dutch. In doing so, the paper is successful as it is clearly written and methodologically no more flawed than the work it is based on. However, the work lacks a critical review of the assumptions made by Kirov and Cotterell (2018). The premise of this and the previous work is that neural networks can be shown to be cognitively plausible models of the learning processes if they seemingly exhibit humanlike learning artefacts such as the micro U-shape learning curve of irregular verbs. Comparing the neural network learning artefacts to the micro U-shape learning curve of irregular verbs observed in children's first language acquisition, however, seems like a needless anthropomorphization as its cause is due to catastrophic forgetting in neural networks observed throughout their training. Specifically, neural networks overlearn examples present in the last couple of batches and forget some of the less recently seen examples. Humans, however, do not tend to periodically forget the past tense of 'hit' after learning a new verb. Neither this nor the previous work justifies why the micro U-shape learning curve of irregular verbs is a salient feature of human learning of past tense learning, while other aspects that are fundamental to neural network training but very distant from how humans learn can be overlooked. For example, in what setting is human vocabulary learning random-type-based instead of most frequently used token-based? Is it really the case that 'bled', 'bred', 'led', 'misled', and 'fled' are as frequently heard and thus as early learned as 'ate' and 'went'? Isn't it the case that some curricula learning should be in place for you to claim that these models are cognitively plausible? If your model does not share any of these aspects in human learning, why should some other phenomena be seen as evidence of their similarity?

Questions:

Your stopping criterion for training is based on a maximum number of training steps. What is a cognitive justification for this? Why not use early stopping on a development set?

A more general question: in what setting humans vocabulary learning is random-type (as opposed to most frequent token-based)? Wouldn't curriculum learning be more realistic?

If so, how comes that the micro U-shape learning curve of irregular verbs is still in any way salient feature of the model?



**Paper Type:**

Long paper

---

### Official Review · Reviewer_1GfZ · 2023-03-10
**The paper indicates that seq2seq models can acquire morphology in the same way as human**

**Rating:** 5
**Confidence:** 1

**Review:**

The paper tries to examine if seq2seq models can acquire morphology of language (specifically past tense) in a similar fashion to how human do. The authors picked a model that works on English and demonstrated the model’s generalizability to Dutch and German. I am not very familiar with the literature to judge if this is a big deal or not. The just affirm the success of a previous model (KC 2018).  While it is interesting to see that this matches the way human learn acquire the language morphology, I do not know if this is a significant contribution.

**Paper Type:**

Long paper

---

### Decision · Program_Chairs · 2023-03-17

Accept